# Chaos-Based Lightweight Cryptographic Algorithm Design and FPGA Implementation

**DOI:** 10.3390/e24111610

**Published:** 2022-11-04

**Authors:** Yerui Guang, Longfei Yu, Wenjie Dong, Ya Wang, Jian Zeng, Jiayu Zhao, Qun Ding

**Affiliations:** 1Electronic Engineering College, Heilongjiang University, Harbin 150080, China; 2Beijing Aerospace Institute of Automatic Control, Beijing 100854, China

**Keywords:** ZUC, lightweight stream cipher, lightweight digital chaos, image encryption, FPGA, NIST, entropy analysis

## Abstract

With the massive application of IoT and sensor technologies, the study of lightweight ciphers has become an important research topic. In this paper, an effective lightweight LZUC (lightweight Zu Chongzhi) cipher based on chaotic system is proposed to improve the traditional ZUC algorithm. In this method, a further algorithm is designed for the process of integrating chaos into the lightweighting of ZUC. For the first time, this design introduces the logistic chaotic system into both the LFSR (linear feedback shift register) and nonlinear F-function of the cryptographic algorithm. The improved LZUC algorithm not only achieves a certain effect in lightweighting, but also has good statistical properties and security of the output sequence. To verify the performance of the LZUC cipher, we performed NIST statistical tests and information entropy analysis on its output key streams and discussed the typical attacks on the algorithm’s resistance to weak key analysis, guess–determination analysis, time–stored data trade-off analysis, and algebraic analysis. In addition, we completed the design of an image security system using the LZUC cipher. Histogram analysis and correlation analysis are used to analyze both plaintext and ciphertext data. At the end of the article, the plaintext and ciphertext images displayed by LCD can be further visualized to verify the encryption effectiveness of the LZUC cipher.

## 1. Introduction

IoT and sensor technologies [1] are currently widely deployed in various domains such as RFID (radio frequency identification) tags, smart cities, smart homes, smart agriculture, industrial automation, security, medical services, and digital consumer. These environments typically consist of sensitive or critical information that needs to be guarded against attacks from outside environments. While their security is a primary concern, the main challenge in providing security for these devices comes from a device environment with limited resources in terms of computing power, memory capacity, chip area, and power consumption, which makes lightweight serial cryptography a major mainstream direction in current serial cryptography research [2,3,4]. Among the various cryptographic primitives, stream ciphers play an important role in protecting the confidentiality of information. Essentially, research on stream ciphers dates back to the early 20th century. In the 1940s, Shannon proved theoretically that OTP (one-at-a-time) ciphers are absolutely secure [5]. Stream ciphers are widely used in both civilian and commercial applications because of their easy hardware implementation and fast encryption and decryption speeds [6]. In the late 20th century, many stream ciphers use a completely linear update function so that their internal states can be viewed as a series of linear feedback shift registers. A typical LFSR-based stream cipher is E0 [7]. However, LFSR is not immune to cryptanalysis, so various nonlinear components are needed to enhance its nonlinearity; these commonly used methods include nonlinear combination and memory combination [8,9].

In 2004, the eSTREAM [10] initiative for stream ciphers was launched in Europe. The eSTREAM project introduced three stream ciphers for hardware applications, which use FSR components. They are Trivium [11], Grain-v1 [12,13,14], and MICKEY-v2 [15]. MICKEY-v2 and Grain-v1 use both NFSR and LFSR. Trivium uses 3 LFSRs, each of which has nonlinear feedback from the other LFSRs in its feedback function. The main idea is that the linear part will guarantee good statistical properties and large cycles, while the nonlinear part will protect against attacks that can target linear cryptosystems [16]. The eSTREAM project has greatly influenced the development of stream ciphers, especially the design and analysis of stream ciphers with maximum internal state and high round initialization. Shortly afterward, byte-oriented stream ciphers SNOW 3G [17] and ZUC [18,19,20] were proposed, and they are both widely used in commercial applications today.

In FSE 2015, Armknecht et al. proposed a new idea (i.e., using keys to design stream ciphers with shorter internal states, not only in initialization but also in key stream generation) and designed Sprout accordingly [21]. Although Sprout was quickly proven to be unsafe [22,23,24], the design idea of Sprout gained attention. The most representative example is that Vahid et al. proposed Fruit [25] and obtained Plantlet [26] by improving Sprout based on this idea. However, the security of the algorithm is difficult to guarantee in all cases because of the improper choice of certain parameters in the design [27]. In 2019, Fruit-80 [28] and Fruit-128 [29] were made public as the final versions of Fruit. The authors claim that, as more easily implemented and secure ultra-lightweight versions, they can resist TMDTO (time–memory data trade-off) attacks while maintaining a smaller internal state. Unfortunately, this complex design approach, which is based on adding new components to existing drive models, goes against the traditional design philosophy of simplicity and ease of analysis, which, in turn, is prone to security risks [30,31].

Chaotic motion is a type of random motion that is unique to nonlinear dynamical systems. Since the parameters and initial state variables of chaotic systems are similar to the keys of traditional cryptography, people started to combine chaos theory with the study of traditional cryptography [32,33,34]. In chaos-based digital communication systems, chaotic signals are one of the most popular problems in recent decades and are widely used in communication security. In 2019, Ding et al. [35] applied a chaotic system to lightweight stream ciphers, proposed a new chaos-based lightweight stream cipher system, and verified the security of the system. In the same year, Rokan et al. [36] constructed a new security system by combining chaotic systems with an improved lightweight AES. The lightweight improvement of AES reduces the processing complexity of the original algorithm and has better statistical performance at the same time. In 2020, Fadhil et al. [37] proposed LAES (a new lightweight advanced encryption standard) based on the combination of chaotic systems by using a combination of logical mappings of different dimensions to construct the S-Box, IP (initial alignment), and other major processes. The results showed that the algorithm could effectively improve the efficiency of encryption of text files. In 2021, Dridi et al. [38] proposed the concept of SPCNG (secure pseudo-chaotic number generators) and proposed a secure system based on multiple chaotic mappings that are well resistant to side-channel attacks and initial vector attacks. Although chaos-based cryptosecurity systems are currently attracting much attention from scholars, there is no shortage of scholars introducing chaos systems into the field of lightweight encryption. However, the designs of these lightweight cryptosystems are mostly based on block ciphers, such as AES or bit-oriented stream ciphers with NFSR structures. The biggest advantage of stream ciphers over block ciphers is that they are easier to implement in hardware. In particular, stream ciphers based on LFSR structures are simpler than those containing NFSRs. It is worth noting that no attempt has been made to apply chaotic systems to byte-oriented stream ciphers based on LFSR structures and implemented in hardware [39,40].

In this paper, we propose an effective lightweight cipher based on chaos and complete the design and FPGA implementation of an image encryption system based on this lightweight algorithm. The specific contributions of this paper are as follows:An effective lightweight ZUC cipher (LZUC) based on chaos is proposed. The stream cipher is based on the three-layer algorithm structure of the traditional ZUC cipher, optimizes the size of the state variables of the LFSR layer registers as appropriate, and proposes a chaos-based iterative scheme of the modulo operation for the problem such that the linearly driven part may not satisfy the modulo addition operation in the prime domain GF(231-1) after the size of the state variables is changed; for the problem of how to introduce a nonlinear function F-layer into the chaotic system, a chaotic sequence selection algorithm is proposed;The lightweight implementation of digital chaotic systems is accomplished through the knowledge of digital circuits. The improvement of the lightweight discrete chaotic system is combined to ensure the safety of the system while achieving the lightweight purpose more conveniently; a scheme that can ensure the accuracy of the chaotic system in the digital implementation is proposed to enhance the degradation resistance of the discrete chaotic system;The above scheme and the modular design of LZUC are completed and the corresponding simulation results are given. NIST statistical tests are performed on the key sequence output from this cipher and its security analyzed in terms of information entropy and resistance to weak key analysis, guessing–determination analysis, time–stored data trade-off analysis, and algebraic analysis of these typical attacks;An image encryption system based on LZUC cipher and four-dimensional hyperchaotic Lorenz system is designed and the hardware implementation of FPGA is completed. The embedded logic analyzer (ILA) of Vivado platform is used to capture the key data in the image encryption system in real time to verify the reliability of the system. The histogram analysis and correlation calculation are used to analyze the encryption effect and security of the image encryption system and further verify the feasibility of the LZUC cipher. Combined with the final LCD display, it can be accurately visualized that the lightweight cipher has good performance.

The structure is arranged as follows: Section 2 introduces the algorithm structure of the LZUC cipher; Section 3 completes the RTL-level design, simulation test, and comparative analysis of hardware resource consumption of the LZUC algorithm; Section 4 performs NIST statistical tests and analyzes the security of the cryptosystem; Section 5 completes the FPGA design and board-level verification of the image encryption system using LZUC algorithm; and Section 6 provides the full text of the summary.

## 2. Lightweight ZUC Algorithm

The LZUC algorithm reduces the bit width of the shift register of the LFSR while retaining the original polynomial and 2^31^-1 operation domain of the traditional ZUC algorithm. Meanwhile, the nonlinear S-box of the F-function is replaced by the discrete chaotic system. The LZUC algorithm takes a 64-bit initial key k and a 64-bit initial vector iv as input, and outputs a 16-bit key stream.

LZUC consists of three main parts: linear feedback shift register (LFSR), bit recombination (BR), and nonlinear function (F). The LFSR of LZUC consists of sixteen 15-bit linear feedback shift registers, which are spliced with the introduced 64-bit chaotic sequence based on the original ZUC structure to ensure the normal feedback of register variables in the 2^31^−1 operation domain. The BR of the LZUC has the same combination as the BR of the ZUC. The only difference is that the register variables S_0H_, S_2L_, S_5H_, S_7L_, S_9H_, S_11L_, S_14L_, and S_15H_ involved in the combination are all 8-bit, which is a change determined by the lightweighting of the 16 shift register variables in the LFSR. In F of LZUC, the 32-bit chaotic sequence C is used to replace the S-box of the original ZUC algorithm, C_L_ and C_H_ are the low 16 bits and high 16 bits of C, respectively, and the remainder of the structure is similar to that of the original ZUC. It is worth noting that the chaotic sequence C is generated by a specially designed chaotic sequence selection algorithm. The structure of LZUC is shown in Figure 1.

### 2.1. Logistic Chaotic Map

The cryptographic properties of chaotic systems, such as initial value sensitivity and pseudo-random properties, fit well with the principles of diffusion and obfuscation in modern cryptography. In addition, the research of chaotic systems in the field of lightweight has gradually gained attention and achieved some results from scholars in recent years. In order to overcome the problem of the limited accuracy of fractional representation in FPGAs, which are devices without embedded operating systems, this section proposes an FPGA processing method that can improve the accuracy of chaotic mapping iterations.

#### 2.1.1. Fixed-Point Decimal Arithmetic Method Based on Hardware

Since FPGAs cannot represent decimals directly, the processing of decimals in FPGAs generally chooses to represent setpoint decimals with integers. To better illustrate the concept of fixed-point decimals, the authors define the format *m*Q*n*: *m* represents the binary bit width used to represent fixed-point decimals, and n bits after the decimal point is called Q*n*. For example, 12 bits after the decimal point is called a fixed-point decimal in Q12 format, and Q0 is what we call an integer. The setpoint decimal *x* is represented by a register variable *y* with bit width *m* (the highest bit of the register is the sign bit), and *n* is the number of binary bits used to represent the decimal in the total bit width of the register; then this setpoint decimal *x* can be represented by an integer *y* in *m*Q*n* format as follows:(1)$$x={\left(-1\right)}^{y[m-1]}y\cdot 2^{-n}$$
where *y*[*m* − 1] denotes the highest bit of the register.

From the above, it is clear that a number in *m*Q*n* format, whether it is an integer or a fixed-point decimal, has the same form of addition and subtraction. However, the multiplication between fixed-point decimals changes the position of *n* in the *m*Q*n* format. In order to ensure the correctness and accuracy of the fixed-point decimal calculation results, for fixed-point decimals in *m*Q*n* format and *a*Q*b* format, we extend the register bit widths representing the two numbers to the same format when performing addition and subtraction operations, i.e., *m* = *a* and *n* = *b*; for multiplication operations, we define register variables of bit-width *m* + *a* to store the results and also extend the binary bit-width of the decimal to *n* + *b*:(2)$$\begin{array}{l} mQn⇄aQb\\ mQn\cdot aQb\to (m+a)Q(n+b) \end{array}$$

Addition, subtraction, and multiplication operations all require expansion of the bit widths of the storage register variables. Then, necessarily, the excess integer and fractional bit widths of the register variables also need to be rounded and saturated with truncation, respectively. Since the actual engineering generally does not allow direct truncation of the extra binary bits used to represent the decimal at the end of the data, this will greatly affect the accuracy of data processing. Therefore, a hardware-based method for rounding and saturation truncation of binary fixed-point decimal data is proposed below:Define the register variable *r* to represent a fixed-point decimal in *m*Q*a* format, and round it to represent a fixed-point decimal in (*m* + 1)Q*b* format with the register variable *r′*. When the highest bit of the register variable *r* is 0:(3)$$r{}^{\prime }[m:0]=\{r[m-1]\|r[m-1:a-b]\}+r[b-1]$$
on the contrary,
(4)$$r{}^{\prime }[m:0]=\{r[m-1]\|r[m-1:a-b]\}+r[b-1]\sum_{i=0}^{b-2}r[i]$$
where *a* > *b*; “‖” in the full text represents a string or byte string concatenator; *r*[*i*:*j*] means to intercept bits *i*~*j* of register variable *r*; *r*[*i*] means to extract bit *i* of register variable *r*.Define the register variable *r′* to represent a fixed-point decimal in (*m* + 1)Q*b* format and use the register variable *r″* to represent a fixed-point decimal in *n*Q*b* format after saturation truncation. When the register variable *r′*[*m*:*n*−1] = (*m* − *n* + 1)′d0 or *r′*[*m*:*n* − 1] = (*m* − *n* + 1)′d(2(*m* − *n* + 1) − 1):(5)$$r{}^{\prime \prime }[n-1:0]=r{}^{\prime }[n-1:0]$$
on the contrary,
(6)$$r{}^{\prime \prime }[n-1:0]=r{}^{\prime }[m-1]\|(n-{1){}^{\prime }d(2}^{n-1}-1)$$
where *i*′d*j* denotes a decimal number *j* represented by a register variable with a binary bit width of *i*.

#### 2.1.2. Lightweight Algorithm for Chaotic System

We choose a logistic chaotic mapping to build a chaotic system with a fixed-point fractional arithmetic precision of 64 bits, and its mathematical expression is:(7)x(n+1)=μx(n)[1−x(n)] μ∈,x(n)∈(0,1].
where we are given the constant μ = 4 and the iterative initial value *x*(0) = 0.2.

From the previous section, (0.2)_d_ is converted to an unsigned binary number in 64Q64 format as *x*_0_ = 64′h 3333_3333_3333_333333. To complete the subtraction calculation of the (1 − *x_n_*) part of the expression, it is common practice to represent the constant 1 also as a binary number in 65Q64 format, and then subtract it from *x*(*n*) after aligning the decimal points. In this case, either the bit expansion of the constant or the subtraction operation will increase the consumption of hardware resources such as registers. Therefore, we use the principle related to the complement code to accomplish this step cleverly. As is well-known, the simplest way to calculate the complement of a negative number is to leave the sign bit unchanged and invert the rest of the bits and add 1. Since all numbers in a chaotic system are 64-bit unsigned decimals, it follows that:(8)$$M=x_{n}+(\sim x_{n})+1{}^{\prime }b1$$
where the symbol “∼“ indicates the inverse by bit, and *M* is the modulus.

Since *x_n_* is a pure decimal, the modulus *M* = 1, so the original equation can be converted to:(9)$$1-x_{n}=x_{n}+1{}^{\prime }b1$$

This approach not only avoids redundant binary bit expansion but also avoids introducing subtractive arithmetic circuits in the digital circuit. For the operation of the *x_n_*(1 − *x_n_*) part of the expression, we use the method mentioned in Section 2.1.1 to store the result of the operation into a register variable in 128Q128 format. From the constant μ = 4, we know that 4*x_n_*(1 − *x_n_*) ∈ (0, 0.25), then bits 127 to 128 are always 0 in the register variable in 128Q128 format. Since in digital circuits, the left shift operation of binary bits is equivalent to the completion of the product operation, we directly intercept bits 63 to 126 of the register variable as the output of bits 65 to 128 of the register, which is the completion of the multiplication by a constant operation. This method avoids the problems of rounding and saturation truncation operations that lead to data accuracy degradation and also reduces a bit-shift operation, which not only improves the calculation accuracy but also reduces the operation steps. The bit-wise processing of data operations is shown in Figure 2.

### 2.2. Lightweight Key Loading

The 64-bit initial key k and the initial vector iv are expanded into sixteen 15-bit sequences as the initial values of the LFSR register cell variable *S_i_*. Let *k* and *iv* be, respectively:(10)$$k_{0}\|k_{1}\|\;\ldots \ldots \;\|k_{15}$$
and
(11)$$iv_{0}\|iv_{1}\|\ldots \ldots \;\|iv_{15}$$
where *k_i_* and *iv_i_* are both 4-bit, 0≤i≤15.

The 128-bit constant D is expanded in the same way as the 16 7-bit sequences:(12)$$d_{0}{\|d}_{1}\|\ldots \ldots {\|d}_{15}$$
where d_0_ = 7′h4b; d_1_ = 7′h17; d_2_ = 7′h39; d_3_ = 7′h72; d_4_ = 7′h69; d_5_ = 7′h3a; d_6_ = 7′h5c; d_7_ = 7′h27; d_8_ = 7′h53; d_9_ = 7′h32; d_10_ = 7′h65; d_11_ = 7′h4d; d_12_ = 7′h74; d_13_ = 7′h1d; d_14_ = 7′h34; and d_15_ = 7′h4e.

For 0≤i≤15, we have:(13)$$S_{i}=k_{i}{\|d}_{i}\|iv_{i}$$

### 2.3. Lightweight Construction of LFSR

Compared with the LFSR layer of the ZUC algorithm, LZUC contains sixteen 15-bit register cell variables *S*_0_, *S*_1_, …, *S*_15_, and the total bit width of the shift registers is lightened from 496 bits to 240 bits. When participating in the iterative update, *S*_0_, *S*_4_, *S*_10_, *S*_13_, and *S*_15_ which are directly involved in the iterative register cells no longer satisfy the condition of mod2^31^-1 operation. Based on this scenario, we cleverly introduce a 64-bit logistic chaotic sequence c and extend these register cell variables to 31 bits each. Apart from that, the remainder of the structure and working mode of LFSR remains the same as the traditional ZUC algorithm. The expansions of the register variables directly involved in the iterations are as follows:(14)$$\begin{array}{l} S_{15}=S_{15}\|C_{4}\\ S_{13}=S_{13}\|S_{16}\left[15:12]\right\|C_{3}\\ S_{10}=S_{10}\|S_{16}\left[11:8]\right\|C_{2}\\ S_{4}=S_{4}\|S_{16}\left[7:4]\right\|C_{1}\\ S_{0}=S_{0}\|S_{16}\left[3:0]\right\|C_{0} \end{array}$$
where *S*_16_ is the new register variable obtained by iteration, and *C_0_~C*_4_ are the partial bits of the chaotic sequence, *C*_0_ = *C* [11:0], *C*_1_ = *C* [23:12], *C*_2_ = *C* [35:24], *C*_3_ = *C* [47:36], and *C*_4_ = *C* [63:48].

### 2.4. Lightweight Nonlinear Function

At FSE 2015, a new idea of using smaller internal states for stream cipher design was proposed, and this idea provides a new direction for the lightweight design of stream ciphers [21]. The F function of the ZUC algorithm is implemented in hardware, and the resource consumption is mainly in the use of registers and the storage of S-box data. The storage of S-box fixed data requires the consumption of more LUT lookup tables. Chaotic systems are widely used in S-box design because of their good randomness and high initial value sensitivity, so we replace the S-box of the F-level in the LZUC algorithm with a chaotic system for improvement. The chaos-based transformation process of the improved nonlinear function F is shown in Algorithm 1, and all the operations in the table are the same as the traditional algorithm, except for Steps 6 and 7. In the algorithm, *C_H_* and *C_L_* is a 16-bit chaotic sequence that injects nonlinear properties to the F-layer. The linear transformation function with *L*_1_ and *L*_2_ is a linear transformation of 16-bit bit-width; “田“ represents mod 2^16^ addition operation, and “⊕“ represents a bit-by-bit “xor” operation.
**Algorithm 1:** Chaos-based transformation process of nonlinear function F1: $F(X_{0},\;X_{1},\;X_{2})$
2: {
3: $W\;=\;(X_{0}⊕R_{1})田R_{2};$
4: $W_{1}\;=\;R_{1}田X_{1};$
5: $W_{2}\;=\;R_{2}田X_{2};$
6: $R_{1}\;=\;C_{H}⊕(L_{1}(W_{1L}\|W_{2H}));$
7: $R_{2}\;=\;C_{L}⊕(L_{2}(W_{2L}\|W_{1H}));$
8: }


Since the linear transformation operation object of the traditional ZUC algorithm is a 32-bit variable, the F function layer due to the lightweight processing leads to the need to complete the linear transformation of *W*_1_ and *W*_2_, which are 16-bit variables. Our approach is to first merge the two 16-bit variables *W*_1_ and *W*_2_, then use the nonlinear transformation algorithm of the traditional ZUC algorithm to improve the nonlinearity of the sequence, and finally intercept the high and low bits and output them separately.

### 2.5. Nonlinear Sequence Extraction Based on Chaos

In Papers 25 and 29, the authors designed a round key function for the Fruit family of lightweight stream ciphers based on the LFSR and NFSR structures. This round key function, in cooperation with a counter, specifies specific bits in the initial key to be involved in the operation of the nonlinear g-function at each clock cycle. Although the proposed Fruit cipher and the corresponding analytical results mark new research progress in the lightweighting of sequence ciphers, the design method of reducing the internal state has been concretely implemented and gradually accepted by the cryptographic community. However, this design approach of continually adding new components and complicating the design based on the existing drive model, such as adding round key functions, adding counters, etc., is questionable. This is because it goes against the traditional design philosophy of simplicity and ease of analysis.

To achieve the purpose of lightweighting the ZUC algorithm without adding additional computing components, the digital chaos system introduced in the LZUC algorithm is applied to both the mod2^31^-1 operation in the LFSR layer and the nonlinear transformation in the F-function layer. This approach controls the bit width of the chaotic system, reduces the resource consumption in hardware implementation, and ensures the proper operation of the iterative operations at the LFSR layer and the nonlinear operations at the F-function layer.

From Section 2.2 we know that all bits of the 64-bit logistic digital chaotic sequence *C* [63:0] is used for the expansion of the state registers *S*_0_, *S*_4_, *S*_10_, *S*_13_, and *S*_15_ in the LFSR. Then, it is a very important question how to select the appropriate chaotic sequence to complete the nonlinear operation in the F-function layer while minimizing the correlation between the mod operation in the LFSR layer and the nonlinear operation in the F-function layer.

In order to avoid the problem of reducing the security of the LZUC algorithm by directly multiplexing the chaotic sequences in the LFSR layer and the F function layer, we intercept the 64-bit chaotic sequence into 8 segments, and the lower 2 bits of each segment are used as the selection bit of that segment of the chaotic sequence, which is used to select the chaotic sequence that participates in the nonlinear operation in the F layer. Consider the chaotic sequence of the *x*th segment as an example: assume that *Cx*[2*x* − 1:2*x* − 2] takes the value of 2′d*y*, then the chaotic sequence selected in this segment is *Cx*[*y* + 4:*y* + 1]. According to this algorithm, a new 32-bit chaotic sequence *C*′ = *C*_7_‖*C*_6_‖*C*_5_‖*C*_4_‖*C*_3_‖*C*_2_‖*C*_1_‖*C*_0_ can be combined, and seq_mux is the 16-bit control register for selecting the chaotic sequence; the extraction algorithm of the chaotic sequence is shown in Figure 3.

### 2.6. Lightweight Bit Reorganization

From the lightweighting of LFSR register cell variables, it can be seen that the input of the BR (bit reorganization) layer changes to 15-bit LFSR register cell variables *S*_0_, *S*_2_, *S*_5_, *S*_7_, *S*_9_, *S*_11_, *S*_14_, and *S*_15_, and the output is four 16-bit words *X*_0_, *X*_1_, *X*_2_, and *X*_3_. This achieves, to some extent, the register resources lightweighting. The calculation process is as follows (Algorithm 2):
**Algorithm 2:** Bit reorganization of LZUC1: BitReconstruction( )
2: {
3: $X_{0}\;=\;s_{15H}\|s_{14L};$
4: $X_{1}\;=\;s_{11L}\|s_{9H};$
5: $X_{2}\;=\;s_{7L}\|s_{5H};$
6: $X_{3}\;=\;s_{2L}\|s_{0H};$
7: }

where “$\cdot_{H}$” denotes taking the highest 8 bits of the sequence and “$\cdot_{L}$” denotes taking the lowest 8 bits of the sequence.

The LZUC algorithm is lightened by the LFSR layer, the BR layer, and the F-function layer, and as can be seen from the above elaboration, the algorithm finally outputs a 16-bit keystream sequence *Z* at each clock cycle as follows:(15)$$Z=W⊕X_{3}$$

## 3. Implementation of LZUC Algorithm

The implementation of the LZUC algorithm was carried out on an Artix-7 (xc7a100t2fgg484-2L). The Visio tool was used for relevant module architecture drawing, Vivado 2019 tool was used for RTL design, and Modelsim 10.7 tool was used for functional and timing simulation.

### 3.1. RTL Design and Verification

At the input of the LZUC algorithm module, it contains a 1-bit clock signal clk, a 1-bit low-level reset signal rst_n, a 1-bit enable signal zuc_en, a 1-bit work start signal zuc_start, a 64-bit initial vector iv, a 64-bit initial key, and a 64-bit key length L. The outputs are a 1-bit keystream output valid flag bit signal and a 16-bit keystream. The top-level design of the algorithm module is shown in Figure 4. The module starts to work after the clock signal is flipped normally, the rst_n signal and zuc_en signal are set to one, and iv, key, and L are given initial values when the algorithm module detects the rising edge trigger signal of the zuc_start signal. The operation mode of the LZUC algorithm includes two types of modes: initialization mode and working mode. The initialization mode first performs 32 rounds of initialization iterations for the LFSR, then enters the working mode, at which time the valid key sequence is output, and the key stream outputs the valid flag position one. The workflow remains the same as that of the traditional ZUC algorithm [41].

The top-level module of the LZUC algorithm consists of SRN (asynchronous reset, synchronous release), ZFSM (algorithm state machine), KE (key extension), CHAOS (logistic chaos generator), LLFSR (lightweight linear feedback shift register), LBR (lightweight bit reassembly), LF (lightweight F-function), and XE (encoding) modules. The top-level architecture of the algorithm and the RTL-level principle are shown in Figure 5 and Figure 6, respectively. The whole cryptosystem is controlled by the control management mode, and the ZFSM module acts as the central controller of the system to control the data interaction processing between each module through precise beat analysis. The state transfer of the ZFSM is shown in Figure 7. The ‘’*’’ in this figure means that the transfer between two states is done unconditionally.

The simulation results of the LZUC algorithm with the initial key k set to 64′h358f_641d_b17b_455b, the initial vector iv set to 64′h1f6b_da6b_fbd8_c766, and the generated key stream length L set to 64′d1024 are shown in Figure 8. The valid key streams are, in order, 16′he5fe, 16′h1e81, and 16′hb6e0, etc.

### 3.2. Hardware Implementation Analysis

The efficiency of hardware implementation of lightweight ciphers is generally measured in terms of circuit area. For FPGA and ASIC implementations, the area is expressed in terms of slice and the number of circuit equivalent gates, respectively [2]. Table 1 gives the resources used for a logistic discrete chaotic system with 64-bit precision implemented on Artix-7 FPGA by the conventional scheme and the lightweight scheme we provided in Section 2.1.

As can be seen from the data in Table 1, the logistic discrete chaotic system implemented with the scheme mentioned in Section 2.1 reduces the resource consumption of LUTs by 38.12% and DSP resources by 20%, respectively, compared with the conventional implementation. Next, the performance of the LZUC algorithm is evaluated in terms of both area and time. Table 2 compares the hardware resource requirements of several lightweight ciphers.

As can be seen from Table 2, due to the inclusion of the chaos module, the LZUC cipher is implemented using relatively more resources, such as LUTs on Artix-7 FPGA. Compared with the Trivium algorithm of the literature [46] and the Grain-128a algorithm of the literature [43], we can see that our internal state is larger. Compared with the Mickey-128 algorithm of the literature [45], the number of LUTs and flip-flop is slightly more, but due to the addition of the chaos module in this algorithm, it improves the algorithm iteration initial value sensitivity, which also reflects the feasibility of adding the chaos module to the lightweight key sequence.

In addition, for a more intuitive resource comparison with LZUC, we also implemented the traditional ZUC algorithm on an Artix-7 FPGA and performed a comparative analysis of the resources of the LUTs of the corresponding algorithmic layers of the two structures. The comparison between the ZUC algorithm and the LZUC algorithm is shown in Table 3. Accordingly, it can be concluded that our proposed LZUC algorithm by improving the traditional ZUC algorithm and introducing a lightweight chaotic system not only improves the nonlinearity of the algorithm, to a certain extent, but also achieves some results at the lightweight level.

The timing delay is also one of the parameters that are crucial to measuring the hardware implementation. The smaller the time delay, the faster the system runs [42]. Table 4 gives the set up delay of the conventional ZUC algorithm and the LZUC algorithm when implemented by Artix-7 FPGA. The total set up delay, logical delay, and netlist delay of LZUC algorithm are 18.212 ns, 8.494 ns, and 9.718 ns, respectively, while the corresponding delays of conventional ZUC algorithm are 16.816 ns, 7.880 ns, and 8.936 ns. It can be seen that our idea of introducing nonlinear chaotic systems in the LZUC algorithm and the designed lightweight scheme do not come at the expense of the speed and performance of the algorithm.

## 4. Key Sequence Analysis of LZUC

This subsection analyzes the output key flow of LZUC algorithm by information entropy calculation and a randomness test. In addition to this, security evaluation is performed for several common attacks.

### 4.1. NIST (National Institute of Standards and Technology) Statistical Tests

The NIST test consists of a total of fifteen tests and is proposed by the National Institute of Standards and Technology. Each test has a different meaning, and these tests allow us to verify the stochastic statistical properties of the data stream [47]. The result of each test is measured by the *p*-value, and if p≥0.01, the test passes. We used the NIST-SP800 test suite version 2.1.2 to obtain a random key stream of length 5 × 10^6^ bits and repeated the random test 100 times with the Linux platform. The test results are shown in Table 5, where the *p*-values of all items in the table satisfy p≥0.01, indicating that the key sequence output by LZUC has good statistical properties.

### 4.2. Information Entropy Analysis

Information entropy is used as a measure of uncertainty for random variables, and this uncertainty is measured by −logP. *P* is the probability that something will happen, and the higher the probability, the lower the uncertainty. The formula of information entropy, which is the expectation of uncertainty, represents the uncertainty of a system. The greater the information entropy, the greater the uncertainty. The random variables under the *X* distribution are independent of each other. Since the key sequence is generated in binary form, the base of the log is taken as 2 in bit. The formula for information entropy is expressed as follows:(16)$$H(m)=-\sum_{i=1}^{N}P(m_{i}{)\log}_{2}P(m_{i})$$
where *N* = 2; $0\leq H\left(m\right)\leq 1$.

The information entropy obtained by counting the probability of occurrence of 0 and 1 in a key stream of length 5 × 10^6^ bits is calculated, as shown in Table 6. Table 7 lists and compares the information entropy of several keys. It can be seen that the uncertainty of the key sequence obtained by the LZUC algorithm is much higher than that of the original logistic chaotic sequence, and the uncertainty is also relatively higher compared with the logic lightweight stream cipher in the literature [35].

### 4.3. Security Evaluation

This subsection will evaluate the security of LZUC ciphers in terms of both weak key analysis and performance against typical attacks.

#### 4.3.1. Weak Key Analysis

The weak key analysis is a common security analysis method for the sequence cipher initialization process [48]. Weak-state weak key refers to the state of the LFSR as an all-0 state after the key is loaded and goes through the initialization process. Assume that the LFSR is in the all-0 state after initialization and two 16-bit memory cells R_1_ and R_2_ of the nonlinear function F take arbitrary values. The attacker can obtain 2^32^ possible initial states by performing an initialization inverse operation on this internal state. Since the LZUC algorithm introduces 112-bit constants in LFSR during key loading, the nonlinear function F introduces 32-bit all-0 values. In this case, the computational complexity for weak key analysis is $2^{32}\times 2^{-144}=2^{-112}$, so it can be considered unlikely that there is a weak-state type weak key.

#### 4.3.2. Guess and Determine Attack

The guess–-determination attack derives other undetermined internal states by guessing a part of the internal state of the algorithm and then combining it with the mathematical relations of the algorithm. LZUC algorithm has 16×15+2×16=272 bits of internal states. If attackers can determine the values of the other 272 m bits by guessing m of them while the values of these 272-bit internal states satisfy an independent uniform distribution, then the attackers need at least (272 m)/16 keywords to build the algebraic equation to determine all the remaining internal states. However, m < 128 is required for a successful guessing determination attack [49]. Therefore, the attackers need to guess at least 9-bit keywords of the LZUC and obtain these 9-bit keywords. It is necessary to obtain the values of memory units R_1_ and R_2_ for at least nine moments. If the attackers use a complex update mechanism to derive the value of the next computation unit from the value of the memory unit at the current moment, they have to guess the values of R_1_ and R_2_ and the inputs X_1_ and X_2_ for a total of 64 bits. Thus, the attackers need to guess at least 576 bits in total. If the attackers guess R_1_ and R_2_ for nine moments directly, then they also need to guess at least 288 bits of internal state in total. The above results show that the LZUC algorithm has a strong ability to resist the guess–determination attack.

#### 4.3.3. Time–Memory Data Trade-Off Attack

The basic idea of the time–storage data trade-off attack is to trade the cost of increased time for decreased space or increased space for the decreased time. From the design principles of USPUK (Ultra-lightweight Stream cipher Permanently Uses Key), it is known that the most necessary condition for a sequence cipher to resist TMDTO attacks is that the size of the internal state is at least greater than or equal to twice its security level [50]. This means that the internal state of the LZUC cipher should be greater than or equal to 2^32^ to effectively resist the classical TMDTO attack. What we can determine is that the space size of LFSR is 2^240^-1. In addition, the space size of the nonlinear F-function is 2^32^. Therefore, the effective internal state of the LZUC algorithm is 272 bits, which is much larger than 32 bits. It can be seen that LZUC can effectively resist TMDTO attacks.

#### 4.3.4. Algebraic Attack

Algebraic attack is a general cryptanalysis method that can be used for security analysis of almost all cryptographic systems. The basic idea is to consider the entire cipher as a system of overdetermined algebraic equations and then use the method of solving a multivariate system of equations to solve the system of algebraic equations to recover the initial key or the corresponding internal state at a given moment [51]. For the LFSR mod 2^31^-1 operation of LZUC, the following system of quadratic equations can be developed: Let x,y,z∈J, z=x+ymodp, $x=x_{30}x_{29}\cdot \cdot \cdot x_{1}x_{0}$, $y=y_{30}y_{29}\cdot \cdot \cdot y_{1}y_{0}$, $z=z_{30}z_{29}\cdot \cdot \cdot z_{1}z_{0}$, denote the i-th bit summation progression by $c_{i+1}$, and let *c*_0_ = *c*_31_, then we have:(17)$$\begin{array}{l} z_{i}=x_{i}⊕y_{i}⊕c_{i}\\ c_{i}+1=x_{i}y_{i}⊕(x_{i}⊕y_{i})c_{i} \end{array}$$

Further, we can obtain
(18)$$x_{i+1}⊕y_{i+1}⊕z_{i+1}=x_{i}y_{i}⊕(x_{i}⊕y_{i})(x_{i}⊕y_{i}⊕z_{i})$$

Multiplying both x_i_ and y_i_ for the left and right sides above, two other quadratic algebraic equations can be obtained. It is easy to verify that these three quadratic algebraic equations are linearly independent. For z=x+ymodp, 45 linearly independent quadratic algebraic equations can be created. The feedback of the LFSR consists of five modulo p additions, so the entire LFSR can be equated to 225 linearly independent quadratic algebraic equations. Similarly, the nonlinear function F can be equated to a linear equation and 45 linearly independent quadratic equations. Assuming that the attacker obtains 18 subkeys, the system of equations involves 16×15+2×16−1+17(5×15+3×16−2)=2328 variables and 45+17(45×5+2×45)=5400 linearly independent quadratic algebraic equations in total. This shows the difficulty of solving the quadratic system of equations. In addition to this, we introduce a nonlinear chaotic system to the system, which greatly increases the degree of difficulty for the attacker to establish and solve the nonlinear equations. Thus, it is difficult for algebraic analysis to perform an effective attack on the system.

## 5. Implementation of Image Encryption System

The previous sections completed the design, implementation, and analysis of the LZUC algorithm. In order to check whether this lightweight cipher can be applied in practical engineering and to observe the encryption effect more intuitively, in this section, we build an image encryption system using LZUC cipher, combined with plaintext correlation technique and chaotic sequence generator. The encryption system supports encryption and decryption of 800 × 480 RGB images.

### 5.1. RTL Design

We have performed the design and board-level verification of the image encryption system using the LZUC algorithm with the help of the same Artix-7 FPGA development platform. The top-level design of the image encryption system and the top-level module interfaces are shown in Figure 9 and Table 8, respectively.

Under the top-level architecture of the system, it contains RGMII_RX (ethernet communication layer), DYN_KEY (dynamic initial key generation layer), ASFIFO_16BIT (cross-clock domain image cache), ENCRYP_IP (cryptographic algorithm layer), AREA_CTRL (multi-resolution image compatibility control layer), IMG_ ENCRYP_FSM (image encryption system control layer), MIG_IP_DRIVER DDR3 (read and write driver layer), ADDRB_CTRL (read key stream address control layer), IMG_ENCODE (image encryption layer), VGA_CTRL (vga control layer), KEY_RAM (asynchronous clock domain key storage layer), and MIG_ FIFO (mig controller read/write cache). Its top-level architecture is shown in Figure 10. The modules filled with gradient colors in the figure indicate that the interaction of data is implemented to transfer across clock domains. The top-level RTL-level view is shown in Figure 11.

The workflow of the image encryption system can be described as follows:At the end of reset, the RGMII_RX module completes the conversion of plaintext data from the RGMII protocol to the GMII protocol. The CMB_16BIT module spells the plaintext data of two adjacent bytes output from the RGMII module into RGB data of RGB565 mode;The rising edge of the cmb_flag signal triggers the DYN_KEY module to start, generating a dynamic initial key and outputting it to the ENCRYP_IP module;The cryptographic core uses this initial key *k* and the *iv* generated by the four-dimensional Lorenz chaos sequence generator to initialize the LZUC lightweight cipher and store the output keystream in KEY_RAM for use. Meanwhile, the ASFIFO _16BIT module completes the transfer of RGB data across the clock domain and stores the image data in DDR3 memory;When the IMG_ENCRYP_FSM module detects that the empty signal is valid, it marks that the image data is completely stored in DDR3 memory at this time. The IMG_ENCRYP_FSM module controls DDR3 to read out the image RGB data after MIG IP processing and then input to VGA module to display the explicit image through LCD;At this time, when the system master controller detects a rising edge of the cipher_disp signal (indicating that the user presses the image encryption button), it will control the IMG_ENCODE encryption module to complete the heterogeneous encryption of the plaintext image and the key stream. The cipher data will be stored in DDR3 memory in real time;When DDR3 finishes storing the whole cipher image data, the IMG_ENCRYP_FSM module then reads the cipher data into the VGA module and finishes displaying the cipher image through the LCD.

The decryption process is conducted under the control of the plain_disp signal, which is not repeated here.

In the image encryption system of this paper, the system consists of three asynchronous clock domains. The operating clocks of the system encryption module and VGA display module are 100 MHz, and the ethernet transmission reference clock is 125 MHz. The operating clock of the encryption core built based on the LZUC lightweight stream cipher is 50 MHz, so the throughput of the LZUC cipher implemented by the Artix-7 series FPGA is 800 Mbps. This result has a similarity with the literature [35] mentioned in several lightweight ciphers with a level of difference. With an adequate key stream for the image encryption system, the encryption system implemented on Artix-7 series FPGAs theoretically needs to use a total of 6.5536 × 10^5^ ns to encrypt a 256 × 256 color image in RGB565 format and a total of 3.2768 × 10^5^ ns to encrypt a 256 × 256 grayscale image in unit8 type. Since most resource-constrained devices generally operate at less than 100 MHz, this encryption speed can better meet the working environment of IoT sensors.

### 5.2. Test and Board Level Verification

A customizable ILA (integrated logic analyzer) IP core can be used to monitor the internal signals of the circuit design. The waveforms of the ethernet pass map, high-dimensional Lorenz sequence generator, dynamic initial key, and LZUC key streams on the capture board via the ILA condition are shown in Figure 12, Figure 13, Figure 14 and Figure 15.

The image encryption system uses an image resolution of 800 × 480. The image encryption and decryption display results are shown in Figure 16, from which it can be seen that the encrypted image shows a snowflake shape and cannot be recognized, and the image can be restored clearly after decryption.

### 5.3. Ciphertext Image Analysis

#### 5.3.1. Histogram Analysis

Figure 17 shows the histogram images of the plaintext R channel, plaintext G channel, plaintext B channel, and their respective corresponding ciphertext histogram images. We can see that the histogram of the plaintext image tumbles, and the histogram of the ciphertext image is very evenly distributed, which can effectively prevent the attacker from obtaining useful information.

#### 5.3.2. Adjacent Pixels Correlation Analysis

Correlation analysis has also been used as a method to test the effectiveness of encryption. The magnitude of the correlation coefficient is another indicator of the algorithm’s resistance to attacks. One of the objectives of image encryption is to make the correlation coefficient between adjacent pixels and their coefficients as close as possible to 0. In this paper, 30,000 pairs of adjacent pixels in each direction are randomly selected from the plaintext and encrypted images for correlation analysis. Figure 18 shows the test results of plaintext and ciphertext images in horizontal, vertical, and diagonal directions.

As can be seen from the figure, the distribution of adjacent pixels in the plaintext image is relatively concentrated, while the distribution in the encrypted image is relatively uniform. To get a more accurate picture of the correlation between pixels in different directions, Equations (19)–(22) are used to calculate the correlation coefficients. The correlation coefficients of the plaintext and ciphertext images are shown in Table 9. The value of the correlation coefficient in the ciphertext image is close to the ideal value of 0, indicating that the correlation between adjacent pixels is much reduced.
(19)$$r_{xy}=\frac{\mathrm{cov}(u,v)}{\sqrt{D(u)}\sqrt{d(v)}}$$
(20)$$\mathrm{cov}=\frac{1}{N}\sum_{i=1}^{N}\left(x_{i}-E(x))(y_{i}-E(y)\right)$$
(21)$$D(x)=\frac{1}{N}\sum_{i=1}^{N}{(x_{i}-E(x))}^{2}$$
(22)$$E(x)=\frac{1}{N}\sum_{i=1}^{N}x_{i}^{2}$$
where *x* and *y* are the grayscale values of two adjacent pixels in the image, cov(*x*,*y*) denotes the covariance, D(*x*) denotes the variance of the variable *x*, and *E*(*x*) denotes the expectation of the variable *x*.

## 6. Conclusions

In this paper, an effective method for lightweighting ZUC based on a chaotic system is proposed for the first time. Based on this, a lightweight digital implementation of the chaotic system is completed, and then an algorithm design for the process of integrating chaos into lightweight is carried out. The digital chaotic system implemented by the lightweight improvement scheme in this paper reduces the resource usage of LUTs by 38.12% and the DSP resource consumption by 20% compared with the conventional design scheme. Lightweight LZUC ciphers use 29.15% and 37.65% fewer LUTs resources in LFSR and nonlinear function F, respectively, compared with traditional ZUC ciphers. The total reduction in LUTs resources required for algorithm layout wiring is 30.41%. From the comparative results of timing delay analysis, it can be concluded that the total delay, logical delay, and netlist delay of the LZUC cipher are slightly better than the traditional ZUC cipher. This indicates that the lightweight chaos-based design scheme in this paper does not sacrifice the speed and performance of the algorithm for the cost. The idea of randomly selecting chaotic sequences to participate in the nonlinear function F operation by setting selection bits in the output chaotic sequences ensures the nonlinearity of the cryptosystem. Due to the introduction of the diffusion principle, it further enhances the statistical properties of the cryptosystem. The NIST test, information entropy analysis, of the output key stream of the LZUC algorithm shows that the lightweight cipher has good complexity and statistical properties. By discussing the typical attacks of algorithm resistance to weak key analysis, guess-and-determine analysis, time–stored data trade-off analysis, and algebraic analysis, it is verified that LZUC ciphers have good security. In addition, we also completed the design of an image encryption system using the LZUC cipher in RGB565 format and completed the board-level verification. The correlation analysis and histogram analysis completed for plaintext and ciphertext data, as well as the board-level LCD results, further visually verified that the LZUC cipher has a good encryption effect.

## Figures and Tables

**Figure 1 entropy-24-01610-f001:**
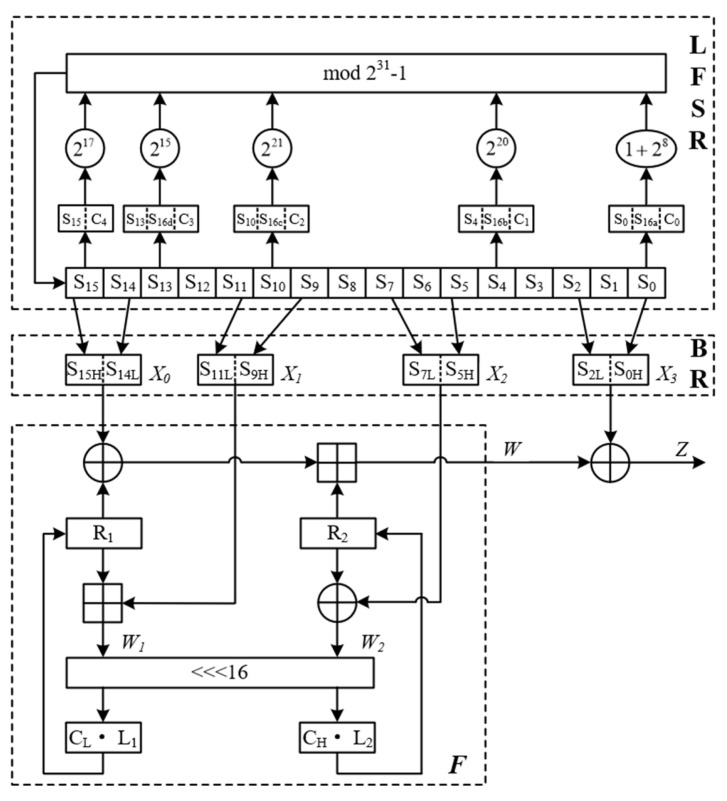
The lightweight ZUC algorithm structure diagram.

**Figure 2 entropy-24-01610-f002:**
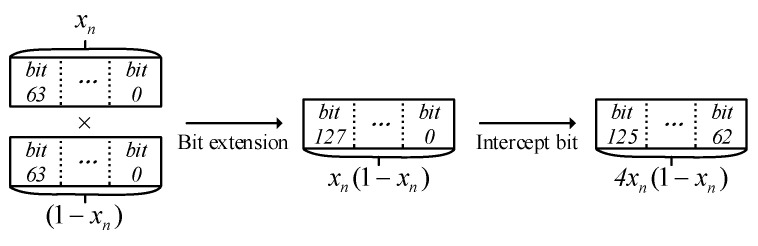
The schematic diagram of data operation bit width.

**Figure 3 entropy-24-01610-f003:**
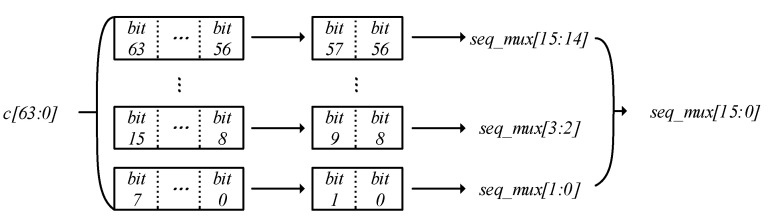
The extraction algorithm of chaotic sequences.

**Figure 4 entropy-24-01610-f004:**
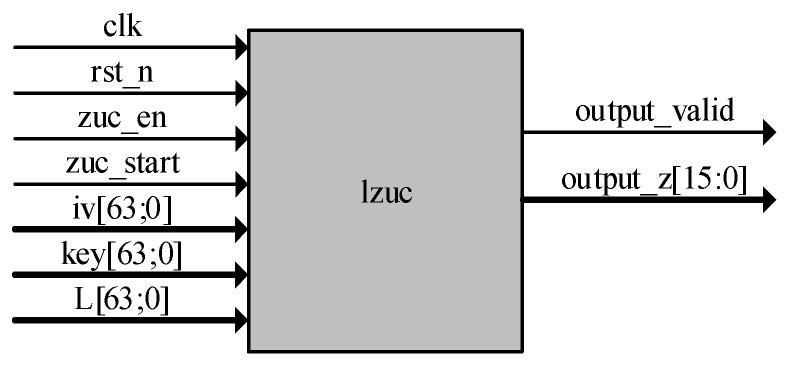
Algorithm module top-level design diagram.

**Figure 5 entropy-24-01610-f005:**
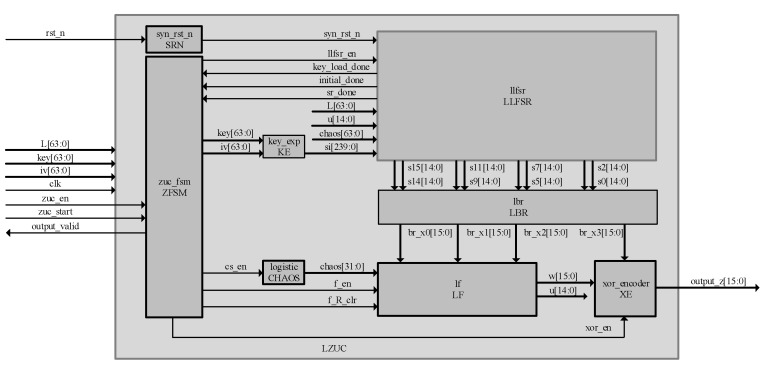
Algorithm top-level architecture diagram.

**Figure 6 entropy-24-01610-f006:**
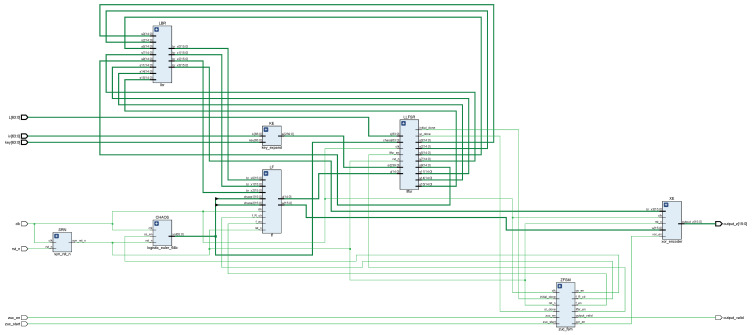
The RTL-level view of the algorithm.

**Figure 7 entropy-24-01610-f007:**
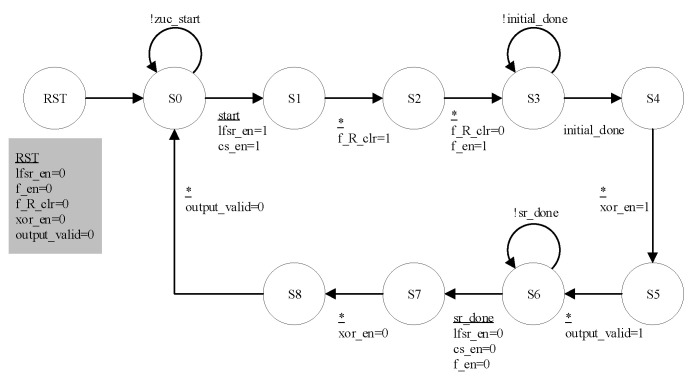
The state transfer diagram of ZFSM.

**Figure 8 entropy-24-01610-f008:**
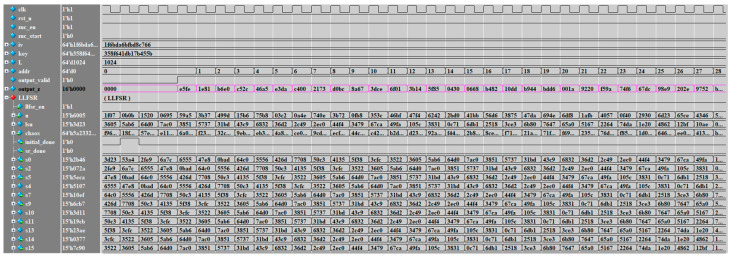
Function Simulation Results.

**Figure 9 entropy-24-01610-f009:**
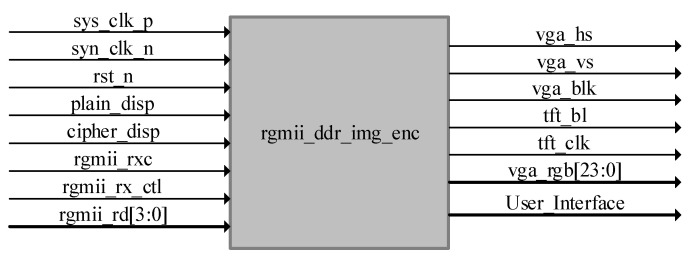
The top-level design of image encryption system.

**Figure 10 entropy-24-01610-f010:**
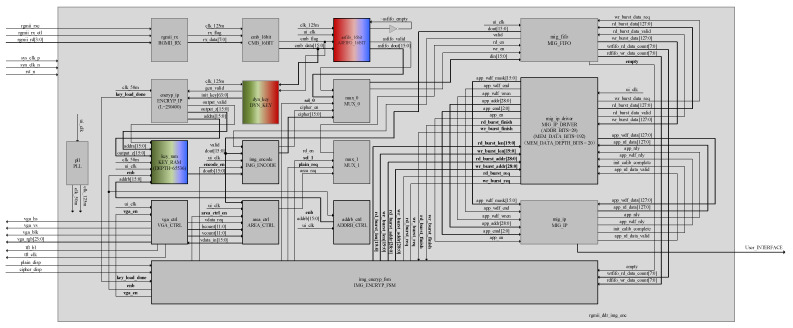
Image encryption system top-level architecture.

**Figure 11 entropy-24-01610-f011:**
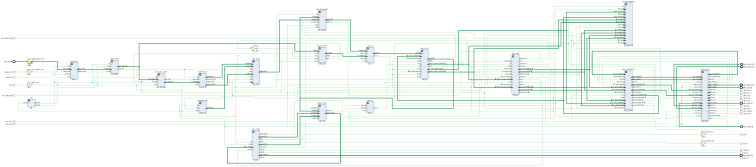
The top-level RTL view of image encryption system.

**Figure 12 entropy-24-01610-f012:**
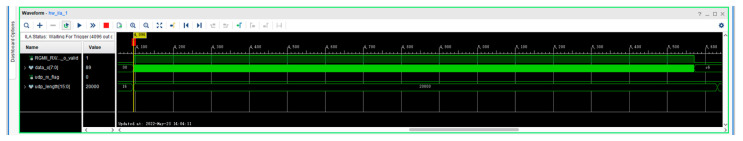
The ethernet transmission ILA capture.

**Figure 13 entropy-24-01610-f013:**
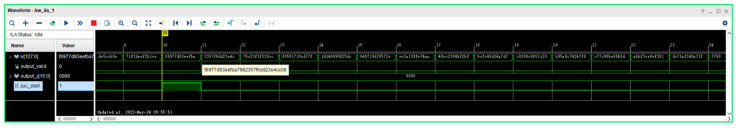
ILA capture based on initial vectors of high-dimensional chaotic sequence.

**Figure 14 entropy-24-01610-f014:**
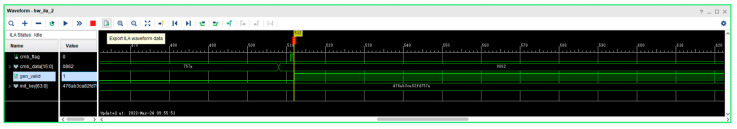
Dynamic initial key generation for ILA capture based on plaintext images.

**Figure 15 entropy-24-01610-f015:**
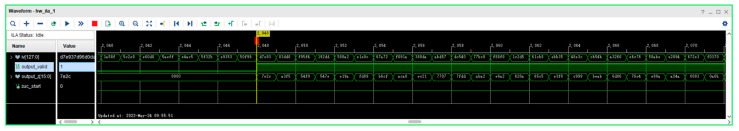
LZUC-based keystream data generation for ILA crawling.

**Figure 16 entropy-24-01610-f016:**
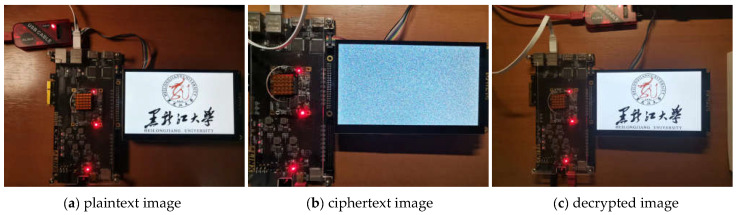
LCD Display.

**Figure 17 entropy-24-01610-f017:**
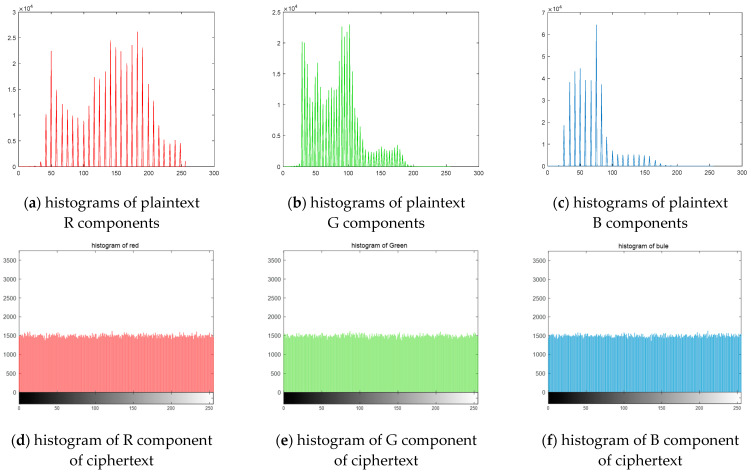
Histograms of plaintext ciphertext images.

**Figure 18 entropy-24-01610-f018:**
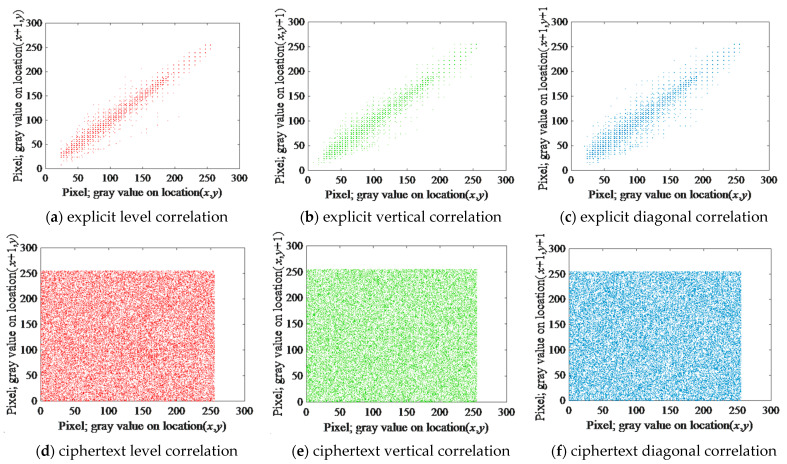
Image encryption and decryption image correlation analysis.

**Table 1 entropy-24-01610-t001:** Comparison of resources before and after logistic discrete chaos lightweighting.

Objects	Slice LUTs	Slice Registers	DSP
Traditional scheme	362	66	20
Lightweight scheme	224	64	16

**Table 2 entropy-24-01610-t002:** Comparison of hardware implementation resources for lightweight passwords.

Cipher	Slice LUTs	Slice Registers	Slice
LAES [42]	9175	-	-
Grain-128a [43]	297	274	-
AES [44]	9516	-	-
Mickey-128 [45]	596	321	200
Trivium [46]	182	289	41
Decim-128	683	330	-
ZUC	960	631	324
LZUC (our work)	668	437	277

**Table 3 entropy-24-01610-t003:** Artix-7 LUTs resources for ZUC algorithm and LZUC algorithm in FPGA.

Cipher	LFSR	BR	F	CHAOS
ZUC	662	0	417	-
LZUC	469	0	39	221
Lightweight proportion	29.15%	0	37.65%	

**Table 4 entropy-24-01610-t004:** Build time delay of ZUC algorithm and LZUC algorithm in Artix-7 FPGA.

Cipher	Total Delay (ns)	Logic Delay (ns)	Net Delay (ns)
ZUC	18.212	8.494	9.718
LZUC	16.816	7.880	8.936

**Table 5 entropy-24-01610-t005:** NIST statistical tests.

Test Project	*p*-Values	Pass Proportion
Approximate Entropy	0.515932	100/100
Block Frequency	0.857405	100/100
Cumulative Sum	0.412490	100/100
FFT	0.063180	100/100
Frequency	0.552568	99/100
Linear Complexity	0.380717	100/100
Longest Run	0.203252	100/100
Non-Overlapping Template	0.857405	99/100
Overlapping Template	0.739918	100/100
Random Excursion (x = −1) Test	0.696376	100/100
Random Excursions Variant (x = 1) Test	0.858470	99/100
Rank	0.627655	100/100
Runs	0.809915	100/100
Serial	0.462821	100/100
Universal	0.223755	100/100

**Table 6 entropy-24-01610-t006:** LZUC outputs the information entropy of the key sequence.

LZUC	Probability P	Information Quantity (Bit)	Information Entropy (Bit)
0	0.499926	1.00021	0.50003
1	0.500074	0.99979	0.49996

**Table 7 entropy-24-01610-t007:** Information entropy of several key sequences.

Cipher	Information Entropy
Logistic	0.5951
Logic [35]	0.9238
LZUC	0.9999

**Table 8 entropy-24-01610-t008:** The top-level signal list.

Signals	I/O	Signal Definition
sys_clk_p/n	In	Differential clock
rst_n	In	Reset, active low level
plain_disp	In	Decryption control
cipher_disp	In	Encrypted control
rgmii_rxc	In	Ethernet receiver clock
rgmii_rx_ctl	In	Valid flag bit for ethernet data
rgmii_rd[3:0]	In	RGMII Input data
vga_hs	Out	Line synchronization signal
vga_vs	Out	Field synchronization signal
vga_blk	Out	Field blanking signal
tft_bl	Out	Backlight control
tft_clk	Out	LCD clock
vga_rgb[23:0]	Out	RGB data
User_Interface	Out	User interface

**Table 9 entropy-24-01610-t009:** Correlation coefficient between plaintext and ciphertext.

Test Object	Color Component	Horizontal Direction	Vertical Direction	Diagonal Direction
Plaintext image	R	0.98567	0.97833	0.96864
	G	0.97411	0.96217	0.94749
	B	0.96279	0.95008	0.93661
Ciphertext image	R	0.00531	−0.00277	0.00120
	G	−0.00869	0.00495	−0.00039
	B	−0.00252	0.00325	−0.00262

## Data Availability

Not applicable.

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
