# Peer review of "Chaos-Based Lightweight Cryptographic Algorithm Design and FPGA Implementation"

_entropy, 2022, doi:10.3390/e24111610_

Round 1

Reviewer 1 Report

The article under review is devoted to the investigation of a chaos-based lightweight cryptographic algorithm and its FPGA implementation. Chaos image encryption is a hot topic today and the manuscript submitted by the authors is a full-fledged study that includes a description of the proposed solution, cryptographic strength analysis, efficiency investigation, and a comparison of the proposed method with other algorithms. The article presents an original new study in the field of the thematic focus of the journal. The lightweight cryptographic algorithm proposed in this manuscript reduces the resource usage of LUTs by 38.12% and the DSP resource consumption by 20% compared to the conventional design scheme.

I have several shortcomings:

  1. Please add a contribution at the end of the introduction section.
  2. Figure 1 is not clear. Please give a wide explanation of the scheme and add a description in empty blocks.
  3. I recommend adding Figure 11 in supplementary material in high resolution.

In general, after appropriate corrections and additions, I approve publication of this manuscript.

Reviewer 2 Report

In this paper, authors proposed a improved lightweight cipher based on chaotic system. The method introduced in this paper has some new ideas in the algorithm design of hardware implementation and has certain application potential. I have the following  comments for reference.

1. First of all, in the introduction, authors should clearly highlight the novelty and contribution of the work, preferably with a bulleted or enumerated list.

2. There many blank squares in Figure 1, some of these boxes should be filled in with the necessary text.

3. For the whole text, variable names should be in italics.

4. In line 143, what is "mQn format"? What does 'Q' mean?

5. What does the 'g' in Eq.(4) and Eq.(18) mean?

6. In line 235, "(represented by 4 and 5 in Eq.)" ,the way it's written is wrong.

7. Eq.(15) should be revised to a Algorithm in a table form. So do Eq.(16).

8. The time cost should be mentioned. How much time does it take to encrypt a 256×256 grayscale image and a color image, respectively, under a specific software and hardware environment?

9. It is suggested that the symbols and languages of the full text should be carefully checked and further revised.

Reviewer 3 Report

Dear authors! Thank you for your interesting and well-structured study. The use of chaos to make encryption algorithms more lightweight is a promising field of research, which is also in demand in IoT. Your work is an important step towards this issue.

Along with the strengths, such as quality of presentation and overall paper design, I would also like to note minor shortcomings of the work.

1. Despite the fact that your algorithm shows very good results both in terms of statistics and in terms of low use of FPGA resources, how would you rate the cryptographic strength of your algorithm in comparison with counterparts from Table 2? After all, Grain-128a is more compact. You need to clearly justify the benefits of your algorithm.

2. Why was the logistic transformation chosen for the cipher? The use of the Chirikov map, and especially the modified Chirikov map would significantly expand the key space.

3. Please label the blocks of the algorithm in Fig. 1.

Use a different font for formulae to make them visually highlighted in text (in Lines 153, 162 etc.).

Round 2

Reviewer 2 Report

The authors have made a great work addressing all my concerns. So, I recommend the paper to be accepted.